# Earthen Construction: Acceptance among Professionals and Experimental Durability Performance

**Juliana F. Nina** [1,*] **, Rute Eires** [2] **and Daniel V. Oliveira** [3]

1 Department of Civil Engineering, University of Minho, 4800-058 Guimarães, Portugal
2 Centre for Territory, Environment and Construction (CTAC), Department of Civil Engineering, University of Minho, 4800-058 Guimarães, Portugal; rute@civil.uminho.pt
3 Institute for Sustainability and Innovation in Structural Engineering (ISISE), Department of Civil Engineering, University of Minho, 4800-058 Guimarães, Portugal; danvco@civil.uminho.pt
* Correspondence: nina.juliana@gmail.com

**Abstract:** Earthen construction is one of the world's oldest and most popular construction methods, and it is still the target of prejudice due to the loss of ancestral knowledge. Due to the need for more effective and healthy building solutions, this study conducted a survey to determine the interest and knowledge of construction professionals regarding sustainable and natural materials and building techniques to understand how open these professionals are to changes in their working methods and if they identify urgency in that change. With the intent of proving the durability of earthen construction materials, laboratory research was developed which involved the preparation and performance evaluation of samples of earthen elements from the most-used techniques: rammed earth and compressed earth blocks. This evaluation was performed using the accelerated erosion test, simulating periods of rainfall and drying, and the post-test loss of resistance was also evaluated. According to the results obtained from the research survey, there is a predominant lack of knowledge about earthen construction and other traditional and sustainable materials. On the other hand, the experiments demonstrated that earthen construction can be durable when using either a small percentage of stabilizing material or a covering plaster.

**Keywords:** earthen construction; natural materials; rammed earth; sustainability; CEB

## 1. Introduction

The world is undergoing significant changes, especially in relation to the climate, which have created an urgent need for a paradigm shift in the conception of construction. This shift requires changes in posture and decision-making towards options that can mitigate the environmental problems caused and provide more effective and sustainable construction solutions while maintaining manageable costs. This urgency for change has been highlighted in recent research on sustainable construction practices [1].

Rammed earth (RE) and compressed earth blocks (CEB) are two of the most popular earthen building techniques and are currently more adapted to current production technologies. RE, known as a traditional building technique, was described by Gilly in 1787 as *"the most advantageous method of building with earth"* [2] with almost no separation between the building technique and the raw material. The main equipment for the execution of the rammed earth technique are the formwork and the compactor, and the technique consists of uniformly compacting slightly moistened soil in successive layers inside the formwork. However, present-day formworks are larger, similar to those used in concrete construction, and the soil processing is also much faster; it is also possible to obtain precast rammed-earth walls [3]. The CEB technique, which is more developed in France, Belgium and Germany, arose from the evolution of adobe and consists of compressing moist earth with a significant percentage of fine particles in formworks, followed by immediate demoulding.

The compression is performed using a manual or hydraulic press, which provides the production of blocks that can be solid or perforated and even small flooring tiles [4–7].

The present study aims to challenge the prejudice regarding the durability of earthen construction, which is often perceived by society as a symbol of poverty, leading to a lack of technical knowledge, investment, or technology among professionals [4,8]. To investigate the durability and efficiency of earthen building techniques, samples of rammed earth (RE) and compressed earth blocks (CEB) were prepared and tested using the accelerated erosion test, which simulates periods of rainfall and drying, followed by an evaluation of the post-test loss of resistance. An ultrasonic analysis was also conducted to assess the homogeneity of the specimens after the durability tests, including the presence of voids or cracks, and to relate it to the compressive strength of the material. This study also involved testing samples with various surface treatments to enhance the durability of earthen materials, including the traditional and natural materials commonly used in earthen construction. This laboratory investigation supports the survey conducted to identify the challenges faced by professionals in the market, including finding qualified labour, equipment, and materials for sustainable construction.

The working methodology addressed: (1) a brief review of ancient earthen buildings to demonstrate how they have remained standing until today, revealing the main benefits of this construction material; (2) the preparation of a survey questionnaire on sustainable and natural materials, including earthen construction, to assess the level of interest and knowledge about this type of construction technique among professionals; (3) experimental testing of RE and CEB samples with different types of binder materials for soil stabilization and several types of finish.

## 2. Earthen Buildings That Stand the Test of Time

Earth was used as a building material by all ancient cultures, from simple dwellings to religious settlements and temples. It is a building material as old as humankind itself: having been used on all continents, it has a universal character. Earthen constructions have been known for over 9000 years. For example, houses made of adobe, built between 8000 and 6000 BC, were discovered in a region of the former Soviet Union; the Great Wall of China, which is approximately 4000 years old, was initially built of adobe and was later covered with bricks and stones; the city of Ouarzazate (Figure 1) in the Draa Valley, Morocco, is approximately 250 years old and is still occupied today [2]. The use of raw earth material has served not only for the construction of rural and urban settlements but also to erect of some of the most valuable and interesting monuments and architectural ensembles in the world, such as the Great Mosque of Djenne in central Mali. With a prayer hall that can hold up to 3000 people, this UNESCO World Heritage site rises almost 20 m high and is equipped with three separate minarets and hundreds of palm trees called "toron" that project from its façade for support and to facilitate plastering in an annual ceremony of rendition [9,10]. It is also noted that almost all African mosques are built with earth. Furthermore, the ancient city of Shibam, Yemen, one of the cities built with earth, is classified as a world heritage site by UNESCO [11,12].

In the Americas, more precisely in Central and South America, adobe construction was familiar to almost all pre-Columbian civilizations. In addition, the RE construction technique was quite popular and was spread by the Spaniards when they arrived in the region [2]. In the medieval period, between the 13th and 17th centuries, the earth was used throughout Europe as filling in timber frame constructions as well as for roofing, by plastering thatched roofs to make them fire resistant. As an example, the historic centre of the city of Rennes, France, has earthen buildings that are more than three centuries old and are still inhabited [5]. The downtown Lisbon area collects buildings from 18th century Portuguese architectural style known for traditional earthen constructions in which a mixture of quicklime, gravel, and hot animal oil with significant water-repellent properties was used as a stabiliser [10].

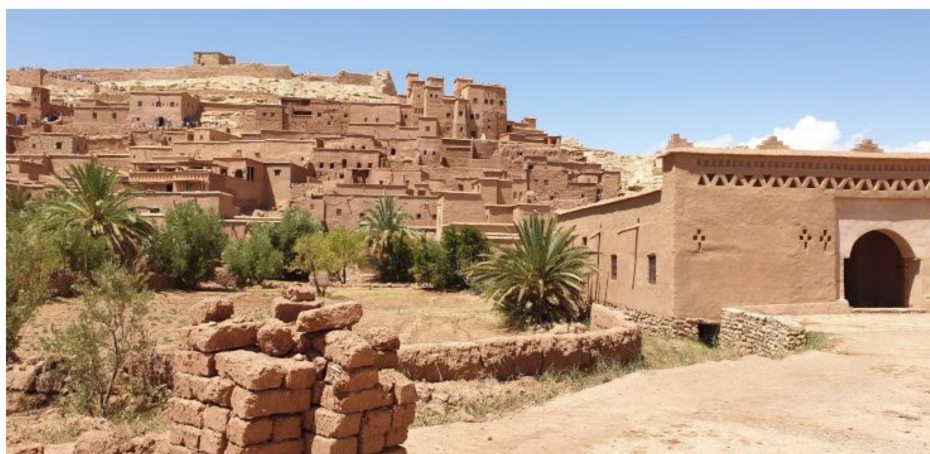

**Figure 1.** The City of Ouarzazate in the Draa Valley, Morocco. (Photo by the author).

## 3. Main Benefits of Earthen Construction

Earthen materials possess properties that make them suitable for constructing structural elements of significant strength, constructing stable and long-lasting building systems, and providing benefits to human health [5]. One property of earthen constructions that positively affects human health is the hygrothermal property, which regulates the humidity and thermal comfort of the indoor environment. Additionally, earthen constructions have a good level of acoustic insulation thanks to their sufficient thickness. Using soil as a building material offers numerous advantages, such as reducing environmental impacts, energy expenditure, being recyclable, and the possibility for reuse in other constructions. Clay, as a non-combustible material, makes the soil fire-resistant [5,13].

In temperate and cold climates, people tend to spend most of their time indoors. Therefore, the indoor environment significantly impacts the health and well-being of its occupants. The feeling of comfort is influenced by temperature, humidity, air movement, irradiation to and from surrounding objects, and indoor air quality. Earthen surfaces can keep humidity levels in balance in enclosed spaces, promoting the well-being of the occupants. An earthen building is known to have excellent hygrothermal behaviour and acoustic efficiency, an absence of volatile organic compound release, and low radioactivity emission rates, depending on the material underneath the construction [1,2]. High indoor humidity levels (RH > 70% during summer) can reduce thermal and respiratory comfort and have a direct adverse impact on fatigue and health. Low humidity levels (RH < 30% in winter) can cause electrostatic discharges, dry skin, and eye irritation [2,10,14–19]. In addition to hygrothermal properties, earthen buildings have other properties that benefit health, such as a good level of acoustic insulation due to thick walls and the rough texture of the material. Earthen buildings also lack the release of volatile compounds present in many construction and decoration materials and have lower emission rates of radioactivity when compared to other building materials [1].

Incorporating natural waste into clay can improve its thermal properties, as demonstrated in a study using coconut powder and sawdust. Increasing the quantity of waste in the mixture decreased the density and consequently decreased the thermal conductivity of the samples [20]. The samples reinforced with coconut dust provided better thermal insulation than those reinforced with sawdust because coconut dust particles are lighter than sawdust particles. Compared to the reference sample, the addition of coconut powder and sawdust resulted in a 26% and 22% decrease in density and a 46% and 43% decrease in thermal conductivity, respectively. Therefore, both types of coconut agglomerate incorporation improved the overall thermal properties of the produced unburned clay blocks [20].

## 4. Assessment of Interest in Earthen Construction and Other Sustainable Materials

A survey was carried out to assess the level of interest and knowledge of professionals regarding earthen construction and other natural materials when compared to the construc-

tion materials commonly used and found in the present-day market. A general survey was prepared so that it could be addressed to a larger number of professionals rather than focusing only on earthen construction, avoiding receiving answers only from those who already knew the material and its techniques. The survey obtained a total of 59 participants and was disseminated to more than 500 contacts between June and September 2021. It was addressed to students and professionals of architecture and engineering from Brazil and Portugal. Thus, the survey sought to determine the profile, motivation, and trend of the current construction market from the point of view of industry professionals to understand the popularity of sustainability in construction. The comprehensive research report, which contains all questions and answers, is available for further examination in [1].

The interviewees were asked about the factors that motivated them to choose a particular construction technique for their projects or designs. They identified durability, maintenance costs, construction techniques and solutions as equally highly relevant factors, followed by the economic factor in second place. Caution with respect to the project and construction time and aesthetic considerations were also considered relevant. In contrast, the use of local materials and low pollution generation/recyclable materials were deemed slightly less important (Figure 2).

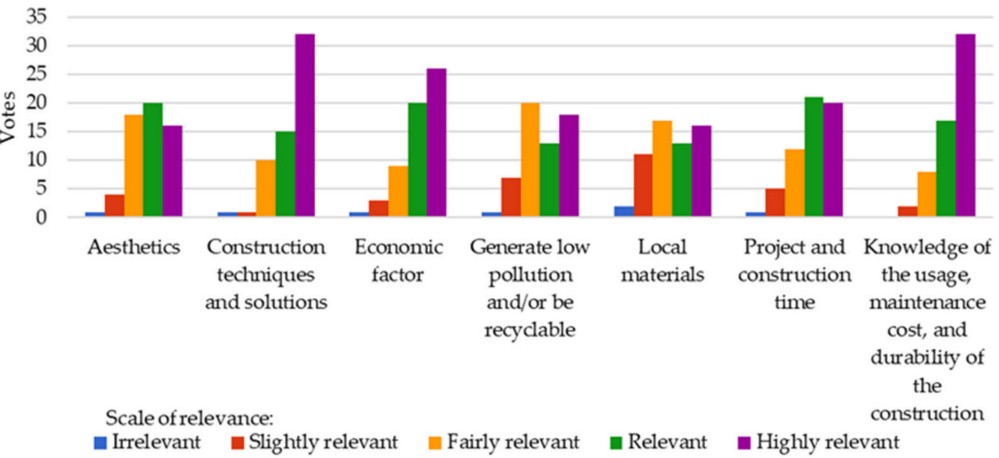

**Figure 2.** Motivations of professionals for choosing a particular construction technique.

The interviewees were also asked about the importance of using sustainable construction practices. Almost half of them (49.2%) stated that having sustainable priorities when designing a new development or rehabilitating an existing one was of extreme importance. Another significant proportion considered it relevant (28.9%), and the remaining proportion considered it to be of medium or low importance. Regarding building rehabilitation, the survey also inquired about the prevalent construction materials, providing options such as conventional and natural/sustainable materials. The findings indicated that earthen and clay mortars, and the External Thermal Insulation Composite System (ETICS) were not commonly utilized options, whereas drywall and cement mortar were the preferred choices among professionals. Medium-relevance alternatives included expanded polystyrene (EPS), lime mortar, and wood (Figure 3).

The study aimed to assess clients' interest, understanding, and acceptance of sustainable building materials. Despite the potential benefits of eco-friendly construction, clients may not be aware of sustainable options and may not initiate requests for sustainable proposals. Therefore, the research explored how frequently professionals present sustainable alternatives to clients. Most respondents reported that they occasionally proposed sustainable options, and the survey also found that clients infrequently (33.9%) or occasionally (44.1%) opted for sustainable materials, with only a small percentage always (1.7%) or often (13.6%) choosing eco-friendly solutions. This finding highlights the need to educate clients and encourage them to prioritise sustainable options. The study also identified the clients' priorities, with economic factors ranking the highest, followed by aesthetic value, project

duration, and construction time. Factors such as maintenance cost, building durability, use of local or recyclable materials, and low environmental impact were considered less relevant to the clients (Figure 4). That a low priority given to such factors may be attributed to a lack of knowledge and awareness, as well as a lack of encouragement from regulatory bodies and the media. Therefore, it is important to raise awareness among clients, professionals, and regulatory bodies regarding the benefits of sustainable building practices.

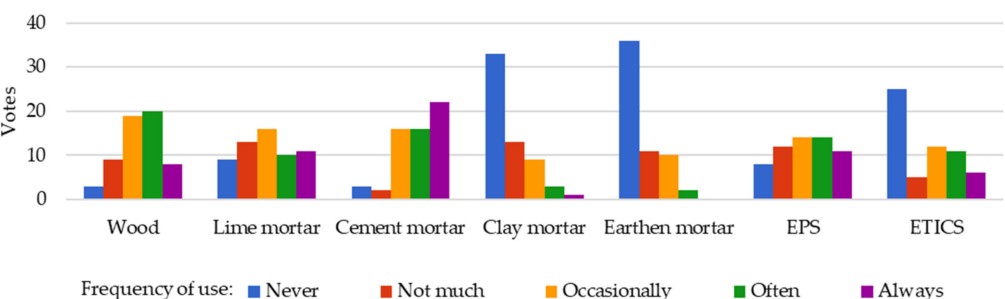

**Figure 3.** Frequency of use of materials in building rehabilitation.

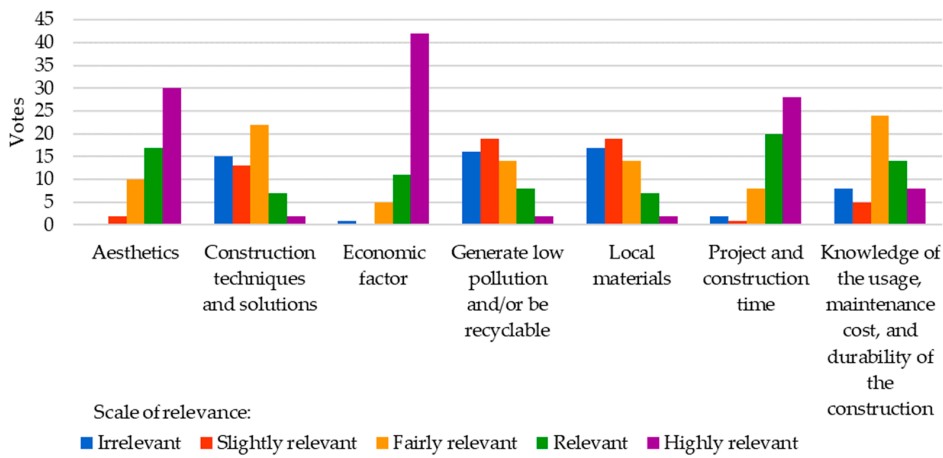

**Figure 4.** Motivation for clients' decision making.

The survey also aimed to identify the challenges faced by professionals in the market, including qualified labour, materials, or equipment for sustainable and natural construction. Interviewees were asked about the main factors that hinder their use of sustainable construction techniques in their projects or works. Finding materials and specialized labour was identified as the most challenging factor, followed by cost and a lack of customer interest. A lack of knowledge, preference for market trends, design, and finish were ranked third, fourth, and fifth, respectively. Only one respondent commented that none of the options would impede choosing a sustainable practice, highlighting the priority of sustainability in the industry. When asked about what they considered lacking for a building to become more sustainable, the cost factor was the main concern, followed by the lack of government incentives and regulatory legislation for sustainable decision making. The need for more information and knowledge about sustainable materials and techniques was also highlighted, as was the importance of convincing clients. Other factors mentioned by participants included the need for collective awareness, safety and guarantees of work, and the increased involvement of colleagues in proposing sustainable solutions to clients (Figure 5).

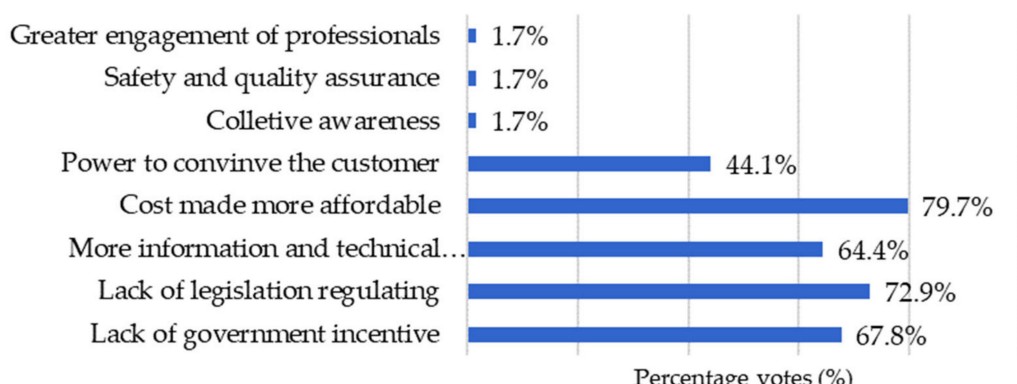

**Figure 5.** Difficulties with the use of sustainable alternatives.

Using the information collected in the survey, it was possible to identify the trends of the professionals surveyed, and it was observed that the factors guiding their steps were most often the same: access to certain types of materials commonly found in the market; the cost of materials and labour; technical knowledge; and the aesthetic factor. Another interesting point was the remarkably low level of responses to the survey. Though the average number of responses to e-mail surveys is around 30% [21], the response rate to this survey was 19.7% (excluding the universities' mailing lists, which provide no perspective of the number of contacts reached). It is assumed that these numbers may be related to the lack of interest in participating in this type of research, as well as the devaluation of the proposed theme by the target audience. Another study published in 2012 researched the marketing and innovation of materials for sustainable construction, using a survey that also addressed construction professionals. This study showed that the sustainable construction market was new and had growth prospects in Europe due to the demand for sustainable building materials that provided energy savings. However, less than 50% were predisposed to invest in sustainable products or services in practice [22]. Therefore, a decade later, it can be noted that there has still not been a mass awareness or explicit change of paradigms regarding sustainable construction.

## 5. Durability Assessment of Earthen Construction

Aiming to understand the feasibility, characteristics, and performance of earthen construction, a study was conducted in the Laboratory of Building Materials, Civil Engineering Department, University of Minho. First, the soil was characterized from the geotechnical perspective to identify its properties and indicate the need for soil correction in order to use it as a building material. Then, the soil was prepared, and a plan was made for the production of CEBs and RE specimens with different binder materials, additions, and coatings in order to evaluate their performance, considering different constructive options. Finally, the specimens were submitted to several durability tests: accelerated erosion by water jet; a visual and qualitative analysis with a pachymeter; an ultrasonic analysis; and a verification of mechanical strength loss by compression.

### 5.1. Materials and Composition of the Specimens

The soil used in this study was obtained from the Minho region in Portugal and was characterised through field and laboratory tests. In the first contact, observation, and field tests, it was possible to notice that it had a very rough and sandy characteristic. With little plasticity, the soil presented a need for correction with clay particles to become more propitious for earthen construction experiments, and 8% kaolin was added to facilitate this process. The granulometric curves of the original soil and the soil with the addition of kaolin can be observed in Figure 6 and are within the ideal parameters for useful soil in construction [22]. A granulometric analysis of the soil by sedimentation was carried out based on the LNEC E-196 standard [23]. The study used a clear, yellowish kaolin which was washed with a hydrocyclone and dried at the factory. According to data provided

by the company Mibal, kaolin has a density that varies between 2.4 and 2.7 g/cm³ and a fineness that is essentially between 2 and 30 μm. To reduce the shrinkage content, the soil used for producing the RE specimens was supplemented with 40% fine gravel (up to 10 mm).

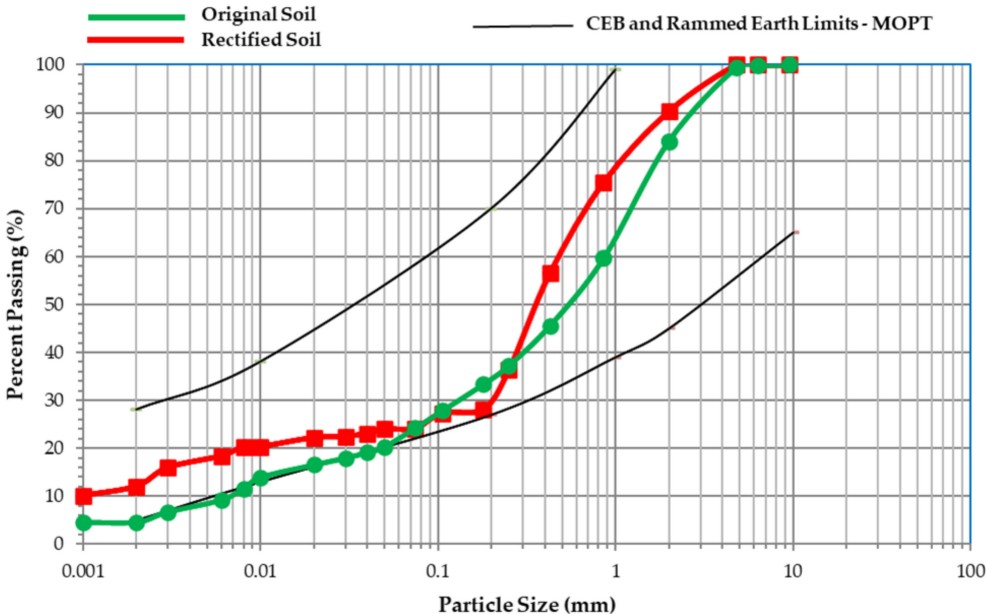

**Figure 6.** Particle size distribution curves of the original soil and the soil corrected with 8% kaolin. (Values of MOPT apud Muguda et al. [24]).

Binding materials such as cement and aerated lime (quicklime and hydrated lime) were individually added to the soil at 6% of the soil mass. These binders are the most common in earthen construction and have the function of increasing the water resistance of the soil, also increasing the compressive strength. The study utilized a limestone Portland cement CEM II/A-L 42.5R, certified in accordance with NP EN 197-1 (2001) [25]. The cement contained 80–94% clinker, 6–20% limestone, and calcium sulfate as a setting regulator. The material's technical data sheet recommends it for soil–cement applications [26]. Hydrated lime CL90 which complied with the European Standard EN 459-1 [27] was used in the study. Micronized quicklime from the brand Maxical was also used.

The CEBs, blocks with dimensions of 22 × 10 × 13.5 cm³, were produced in a manual press. The rammed-earth specimens were produced using a pneumatic compactor and a wooden pestle with which the earth was compacted in 5 cm layers inside iron forms, forming 15 × 15 × 15 cm³ blocks. The soil hydration of all specimens was up to 14% of the soil volume, and in the case of the addition of quicklime, more than half of its volume of water was added for previous hydration. For the mixture of soil with quicklime, the old method of hot hydration was used, that is, 48 h before the preparation of the specimens, alternating layers of soil and lime with water irrigation were placed in a container [2,10]. The curing period was 30 days at room temperature (approximately 20 °C and 60% RH).

Specimens without addition and specimens with different types of finish were also produced to evaluate the durability of earthen construction under different protection conditions. These were also used to evaluate the quality of the surface finish on the specimens as healthier and more natural and affordable alternatives. Part of the studied specimens, those without addition or with lime addition (quick or hydrated), received various finishes: lime mortars with and without hemp fibres, used to aid adhesion and reduce shrinkage damage; lime paints with pigments; *aloe vera* paint [28]; olive oil soap, reproducing part of the Moroccan technique of *tadelakt* [29]; cooking oil already used with caustic soda [10]; and handmade varnish, which included beeswax and pine resin among its components. Further information about all the materials used in the specimens can be

found in [1]. Table 1 presents the composition of each specimen, noting that only one of each type was subjected to erosion and compressive strength testing. CEBs 2, 3, and 7 were specifically chosen for compressive strength testing to enable a meaningful comparison with similar specimens.

**Table 1.** Specimens produced.

| Compressed Earth Blocks—CEBs | | | RE Specimens | | | |
|---|---|---|---|---|---|---|
| **Identification** | **Stabilization** | **Finishing** | **Identification** | **Stabilization** | **Finishing** | |
| | | | | | **Face A** | **Face B** |
| CEB 1 | cement | - | RE 1 | cement | - | - |
| CEB 2 | cement | - | RE 2 | cement | - | - |
| CEB 3 | hydrated lime | - | RE 3 | hydrated lime | - | - |
| CEB 4 | hydrated lime | - | RE 4 | hydrated lime | - | - |
| CEB 5 | quicklime | traditional varnish | RE 5 | quicklime | - | - |
| CEB 6 | quicklime | - | RE 6 | quicklime | - | - |
| CEB 7 | quicklime | - | RE 7 | no addition | - | - |
| CEB 8 | quicklime | soaping | RE 8 | no addition | - | - |
| CEB 9 | quicklime | quicklime/aloe paint | RE 9 | no addition | w/QL plaster | w/HL plaster |
| | | | RE 10 | no addition | w/QL plaster + pigmented lime paint | w/QL plaster + oil painting |
| | | | RE 11 | no addition | w/HL plaster + pigmented lime paint | w/HL plaster + oil painting |
| | | | RE 12 | no addition | w/HL plaster + traditional varnish | traditional varnish |
| | | | RE 13 | quicklime | soaping | - |
| | | | RE 14 | quicklime | quicklime/aloe painting | - |

The CEBs 1, 3, and 5 were made simulating mini walls using two CEBs of each type of addition, cement, hydrated lime and quicklime; this latter (CEB 5) still received a traditional varnish paint finish. The CEBs were joined using 170 mm spread mortars prepared with the respective additives (cement, hydrated lime, or quicklime) in accordance with the EN 1015-3 standard (1999) [30]. These mortars were also reinforced with hemp fibres to improve adhesion and reduce shrinkage damage. The specimens were tested in a coupled format solely for the erosion test. Subsequently, compressive strength testing was performed on the specimens, with the specimens evaluated separately.

For some RE specimens, finishing mortars of quicklime and hydrated lime of 160 mm spread were made with and without hemp fibres to be applied on the surfaces of some specimens of quicklime and hydrated lime. Some surfaces were also painted for finishing and protective purposes. The specimens were subjected to a water erosion test and then tested mechanically under compression.

### 5.2. Performed Tests

The CEB and RE specimens were tested mainly for their durability with the accelerated erosion by water jet test, alternated with drying periods to simulate rain. Complemen-

tary tests were also performed, such as a visual qualitative evaluation and dimensional evaluation using a pachymeter and mechanical resistance by compression.

### 5.2.1. Accelerated Erosion

This test is based on the ASTM D559 [31] standard and the method developed by Rezende [32] to evaluate an accelerated degradation test rain simulator. At 60 days of curing, the samples were placed in an oven at 70 °C for an initial drying period. They were then weighed and left at an ambient temperature for a few hours to cool down. The rain simulation was then initiated, alternating with drying periods of at least 14 h at 70 °C in an oven. During the rain simulation, the specimens were positioned on a 50 mm × 50 mm stainless steel grid platform that allowed the water to fall (Figure 7). The wetting and drying cycles were repeated for three consecutive days. The water erosion test simulated periods of rain, considering the highest average annual rainfall in Portugal and corresponding to an exposure time to weathering of between 43 and 64 years, depending on the type of specimen and distance from the water jet, according to the methodology of Rezende et al. [32]. Therefore, the longevity level of the two studied construction techniques can be estimated.

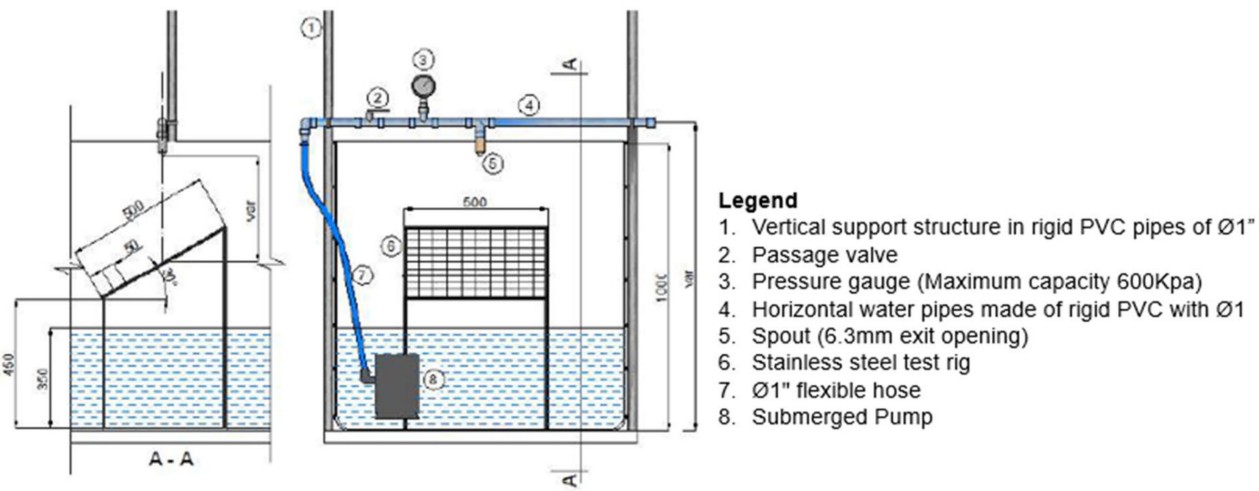

**Figure 7.** Rain simulating equipment (mm) (image source: Rezende et al. [32]).

The sample was subjected to a flow rate of 14.26 litres/min and a water outlet pressure at the nozzle of 45 kPa, which is consistent with the pressure values for erosion tests referenced in international standards [32], see also Figure 8. The results were analysed after three cycles, and the mass losses and changes in the water volume of the specimens were verified. The mass variation was checked at the end of each cycle, and the ratio between the mass of the specimen after the first and third cycles was used to evaluate the change in mass of the specimens. The result was computed based on the NBR 13554 standard [33]. Following each rain simulation cycle to which the specimens were subjected, a visual evaluation was performed to document the changes.

### 5.2.2. Ultrasonic Velocity

This test was conducted to measure the speed of wave propagation through the compacted or compressed earth. It was intended to obtain information regarding the homogeneity of specimens after durability tests, to evaluate the presence of voids or the existence of cracks, and to relate the obtained values to the compressive strength values of the material. An initial ultrasonic evaluation was performed on each group of specimens before they were subjected to rain and drying simulation cycles. Subsequently, the same ultrasonic evaluation was performed to determine what internal changes specimens might have undergone after the three durability test cycles to which they were submitted.

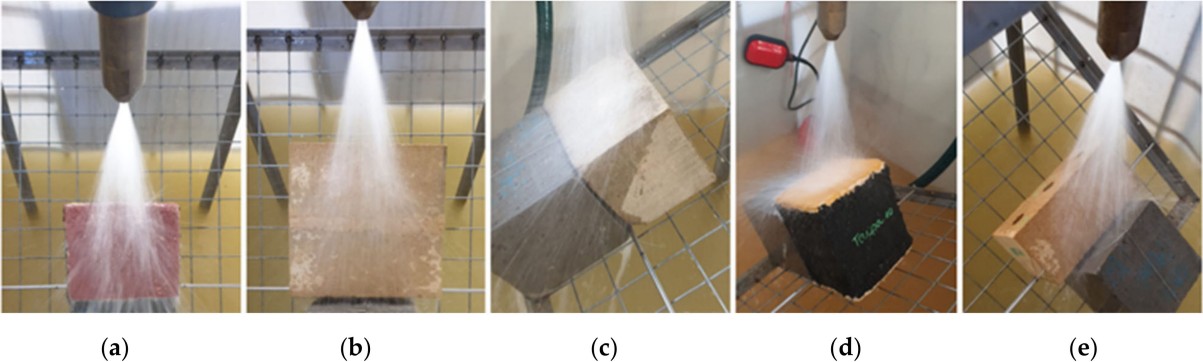

**Figure 8.** Testing specimens of different types: (**a**) RE specimen with mortar and paint; (**b**) CEBs united; (**c**) RE specimen without finishing; (**d**) RE specimen with two-sided plaster and paint; (**e**) CEB.

The direct reading method was followed to obtain results with greater accuracy because the distance between the transducers is more precise, ensuring linearity, and the maximum vibration energy is transmitted perpendicular to the face of the transducers, thus providing more reliable travel times.

The equipment used was the *Pundit Lab + Ultrasonic Instrument* from PROCEQ. With cylindrical transducers, a mass was used as a conductor between the transducers and the specimens' surfaces. Prior to the beginning of the measurement, it was always necessary to calibrate the transducers so that they had a calibration of 25.4 µs. The frequency used was 54 kHz, the length to be measured and adjusted before each measurement [34]. The testing process was based on EN 12504-4 [35], and Figure 9 shows the reference points for the measurement of the RE and CEB specimens.

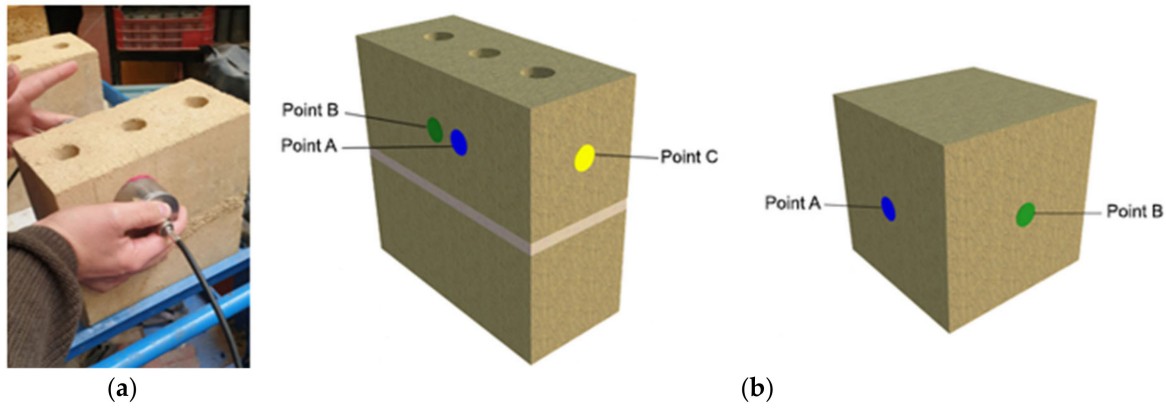

**Figure 9.** Ultrasonic test: (**a**) equipment, transducers and specimen; (**b**) measurement reference points in each specimen typology.

As direct measurements of the specimens were made, the transducers were positioned face-to-face on opposite sides of the specimens, and each measurement was performed three times and then averaged. The readings were made before the durability tests and at the end of the cycles. The measuring distances are present in Table 2.

**Table 2.** Distance to be measured in each typology.

| Measuring Distance (mm) | | |
|---|---|---|
| Points | CEB | RE specimen |
| A | 105 | 150 |
| B | 105 | 150 |
| C | 220 | - |

### 5.2.3. Compressive Strength

The compressive strength test was performed based on EN 772 [36] to evaluate the compressive strength of the CEB and RE specimens. With this test, it was possible to compare the strength of the samples submitted to the durability test against the result of the strength of the samples that were not submitted to the test and that had the same characteristics and type of addition.

As the study also tested CEB specimens joined by mortar in the erosion test (CEBs 1, 3 and 5), the blocks that remained attached during the erosion test were disconnected for the compressive strength test. The upper units of the connected blocks that demonstrated the most significant changes during the erosion test were selected for the compressive strength evaluation. The mortar was then completely removed from these blocks before testing to ensure that all specimens could be compared equally for compression strength testing. Moreover, it is recommended that for earth blocks with nominal heights greater than 71 mm, compressive strength should be evaluated separately for the entire block, in accordance with the references [37,38].

Another study in 2018 investigated experimental procedures to propose suitable testing methods for adoption as standards [39]. It concluded that a slenderness ratio of two would be optimal for testing soil mechanics, resulting in accurate measurements of friction (due to confinement from the press platen), the gradient of dry density, and volume (including gravel or small stones) in earth samples. It was emphasised that compression tests on samples with a smaller slenderness ratio would yield unrealistic results [40]. However, while producing samples with a slenderness ratio of two is relatively easy for rammed earth or cob, it is more challenging for CEBs and adobe. While some proposals have emerged, including overlapping blocks joined by mortar [41], no consensus has been reached [39]. The DIN 18945 standard states that for earth blocks with a nominal height ≤ 71 mm, the compressive strength test should be carried out on two half-blocks stacked on top of each other and connected by a cement mortar [37,38].

To perform the test, a monotonic compressive load was uniformly distributed and continuously incremented until rupture after 1 min. The direction of compression applied to the specimens was perpendicular to the specimen faces that were tested with water exposure and where the finishes were applied, acting in the same direction as the models were compressed or compacted, thus simulating the load that the material would suffer in real construction.

### 5.3. Test Results

### 5.3.1. Accelerated Erosion

During the cycles, all visual changes suffered by the specimens were observed, such as colour change, the appearance of or increase in cracks or holes, a change in surface texture, and the loss of protective finishing. Regarding mass loss, some non-stabilised samples suffered a significant mass loss in the erosion test, resulting one of the non-stabilised rammed earth specimens losing more than 42% of its mass in 15 min of exposure to water. However, specimens with stabilizing materials or with coating mortars obtained lower variation values that were almost null (Figure 10).

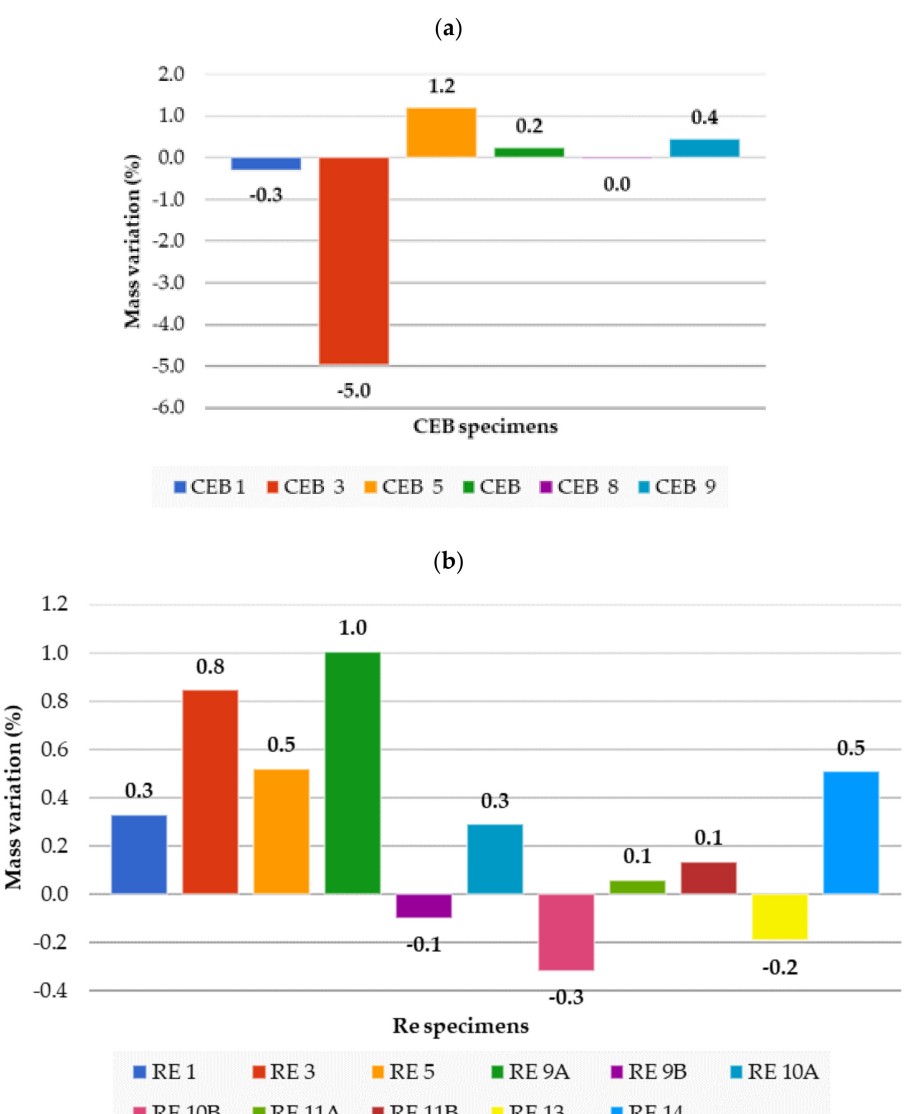

**Figure 10.** Mass variation: (**a**) CEB specimens; (**b**) RE specimens.

Depending on the composition of the specimens, some showed no signs of erosion and even increased in mass. However, with the addition of hydrated, the specimen CEB 3 experienced a 5% decrease in mass, which suggests a potential material loss or an increase in voids. Despite this loss, there were no visual indications of mass loss. The study by Guettala [10] on CEB revealed that there is a mass loss of 1% in soils with the addition of 5% cement and a mass loss of 2% in soils with the addition of 8% lime, which compares well with the results obtained in this study, see Figure 6. One factor that masks the mass loss is the drying period, which may not be sufficiently long enough to completely evaporate the water contained in the specimens at the time of measurement. Repeating the tests with specimens of the same type can also help to identify the variation trend of each type of composition.

The results of the visual and pachymeter surveys of selected specimens subjected to wetting and drying cycles during a rain simulation test are presented in Tables 3 and 4. These tables describe and quantify the observed degradation of each specimen, providing valuable insights into the potential damage caused by the test.

**Table 3.** Qualitative evaluation of CEB specimens (A = Specimens and B = Parameters: 1—Visual register; 2—Evaluation; 3—Degradation dimensioning).

| Qualitative Visual and Pachymeter Evaluation | | | | |
|---|---|---|---|---|
| Wetting and Drying Cycles | | | | |
| A | B | 1st Cycle | 2nd Cycle | 3rd Cycle |
| CEB 1 | 1 |  |  |  |
| | 2 | No relevant changes were observed in the specimen body, and the overall dimensions of the model also remained the same. | | |
| | 3 | Insignificant or non-existent | Insignificant or non-existent | Insignificant or non-existent |
| CEB 3 | 1 |  |  |  |
| | 2 | Material loss occurred mainly near the laying plaster and on existing vertical marks. The specimen's edges had pre-existing material losses. | There was a slight increase in the degradation previously reported, especially around the laying plaster, which has been gradually disintegrating since the last cycle. | The CEBs in the sample separated because the laying plaster that held them together weakened. The analysis focused on the CEB at the top of the sample, which exhibited more changes during testing. |
| | 3 | - | The overall dimensions are approximately 90mm long and just over 10mm deep. | - |
| CEB 9 | 1 |  |  |  |
| | 2 | The specimen has experienced gradual paint loss with superficial wear, a small section of complete detachment, and yellowed areas of paint. | No significant degradation reported from the previous cycle. | There has been no progression of surface degradation since the previous cycle. |
| | 3 | The paint loss is almost 16mm long, with a depth of just over 1mm. The other changes of the exposed face spread to practically the entire surface. | | |

**Table 4.** Qualitative evaluation of RE specimens (A = Specimens and B = Parameters: 1—Visual register; 2—Evaluation; 3—Degradation dimensioning).

| Qualitative Visual and Pachymeter Evaluation | | | | |
|---|---|---|---|---|
| Wetting and Drying Cycles | | | | |
| A | B | 1st Cycle | 2nd Cycle | 3rd Cycle |
| RE 1 | 1 |  |  |  |
| | 2 | The specimen showed no noticeable changes during the cycles. | | |
| | 3 | Insignificant or non-existent | Insignificant or non-existent | Insignificant or non-existent |
| RE 3 | 1 |  |  |  |
| | 2 | The specimen experienced mass loss and fine aggregate exposure in areas where roughness existed from its manufacturing. | Small increase in degradation in the same areas already affected by water in the previous cycle. | Small increase in length and depth of areas already worn during cycles. |
| | 3 | Approximate maximum damage dimensions: 6.00 mm long and 3.50 mm deep. | - | The overall dimensions are 91.86 mm long and 4.46 mm deep. |
| RE 8 | 1 |  | - | - |
| | 2 | Loss of much of the volume in 15 min of water jet exposure. | - | - |
| | 3 | Loss of almost 50% of volume | - | - |

## 5.3.2. Ultrasonic Velocity

Figure 11 shows the average of the three ultrasonic readings taken before and after the cycles. CEB 1, which received cement additions, demonstrated a velocity significantly higher than the other CEBs with lime additions. This difference seems to indicate that the

presence of cement reduces the voids inside the sample, but more tests would be needed to prove this.

| | CEB 1 | CEB 3 | CEB 5 | CEB 6 | CEB 8 | CEB 9 |
|---|---|---|---|---|---|---|
| ■ Pre-cycle measurement | 2261 | 1394 | 1492 | 1453 | 1351 | 1417 |
| ■ Post-cycle measurement | 2323 | 1608 | 1801 | 1570 | 1547 | 1597 |

**Figure 11.** Comparison of the average variation of CEBs before first and third cycles.

Concerning the variation in wave velocity in the CEBs before and after the cycles, it is possible to observe that this variation is always positive. In the case of cement, the presence of water increased the formation of solid crystals and promoted the cohesion of the material. In the case of quicklime, the hardening was due to carbonation, which can also be increased in the presence of water since water carries $CO_2$, also promoting the cohesion of the material. However, it takes a long time due to the slow carbonation process, which justifies the higher values since the first measurement for specimens with cement at the test age (between 60 and 70 days). Another study using specimens stabilised with lime stated that the honeycomb structures can be broken up but will agglomerate again and form more entangled structures that improve cohesion; although they have more voids, they have less bonding between them [42].

It can also be seen that the specimens with added lime obtained greater variations before and after the cycles. CEB 5, with the addition of quicklime and traditional varnish painting, was demonstrated the greatest variation of more than 20%. This may be because the drying time was insufficient for this specimen due to the varnish layer. It was observed during the tests that this specimen created an area of vapour between its base and the surface where it rested during the ultrasonic measurements. Among the CEBs with a lime addition, CEB 6, which had a lime addition and was without finishing, presented the lowest variations in wave velocity.

Figure 12 shows the results of the ultrasonic analysis performed on the RE samples. Here, a similar behaviour to the CEB samples was found, but with lower variation values. The results show that RE1, with the addition of cement, had a higher velocity and was very expressive concerning the other lime addition samples, confirming the trends already discussed for CEB samples.

The RE 3 specimen showed the highest variation (the addition of hydrated lime and without mortar). The washing of the hydrated lime and fines likely caused a considerable readjustment of the materials, with a greater chemical bond between the additive and the soil, although more characterization tests would be necessary to prove this reaction. The RE 1 sample (the addition of cement) varied the least, with a very small difference concerning the RE 13 sample of only just over 1%, followed by the sample of RE 14, with which it also maintained a small difference of almost 2%. The RE samples that received different finishes on more than one face did not have their ultrasonic measurements checked because the result of the ultrasound analysis would be inconclusive since the specimen would be influenced by the tests previously performed on one side with a different finish.

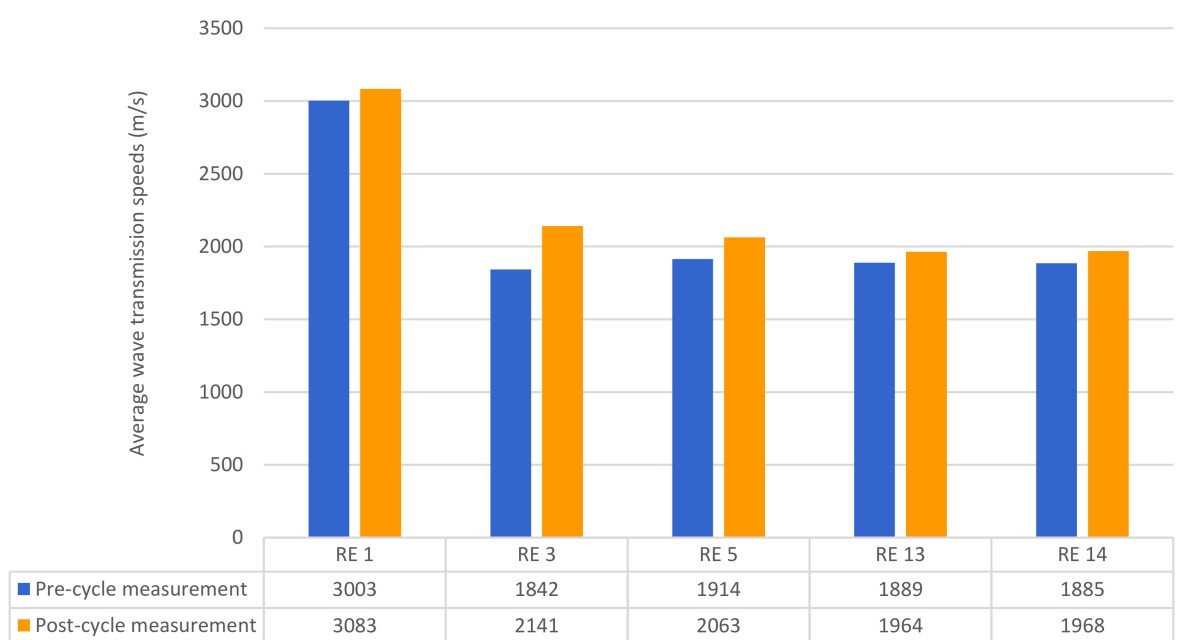

**Figure 12.** Comparison of the average variation of RE specimens between first and second cycles.

### 5.3.3. Compressive Strength

All samples were tested under compression, including those that were not submitted to the erosion test, so that the results could be compared with the samples submitted to the erosion test. A study that tested lime-enriched, red-bed soil specimens showed that the greater the amount of lime and exposure to wetting and drying cycles, the more the specimens were compression resistant and stable [42]. The particle size and strength of consolidation are significant for soil strength [43]. The addition of lime leads to a series of changes to the soil through chemical reactions. The products of these reactions bind the surrounding soil particles together and significantly strengthen the soil [44,45].

The samples in blue colours represent those with cement additions, which presented higher compressive strength results, as expected. The dark blue specimens are the ones previously submitted to the erosion test. In the case of the CEBs, water-exposed CEBs presented higher strengths than their equivalents that were not exposed to water, because the action of water on the specimens, as a cohesion factor of the composite materials, caused an improvement in their performance in terms of compressive strength. The same increase occurred with the samples of RE that had additions of hydrated lime and quicklime. On the other hand, the RE specimens with the addition of cement showed a slightly lower strength limit than similar specimens that were not exposed to water, and the same occurred for the CEBs with an addition of hydrated lime (Figure 13).

The specimens with different surface finishes can only be compared with those without finishes. The green coloured CEBs with the addition of quicklime and protective finishes can be compared with CEB 6 and CEB 7 (without finish), which obtained equivalent results. In the RE results, the light green specimens with different finishes and without stabilizer can be compared with specimen 7 (without stabilizer), though it was not exposed to the rain test. These specimens (RE 9, 10 and 11) have lower resistances and the lowest of all specimens produced, which can be explained by the fact that these specimens had twice the exposure time to water action than the rest of the RE specimens since they had two sides with different plasters and finishes which were tested on separate occasions. The purple-coloured specimens, with added quicklime and paints, showed results equivalent to the specimens RE 5 and 6, with the same type of stabilizer and without protective finishes.

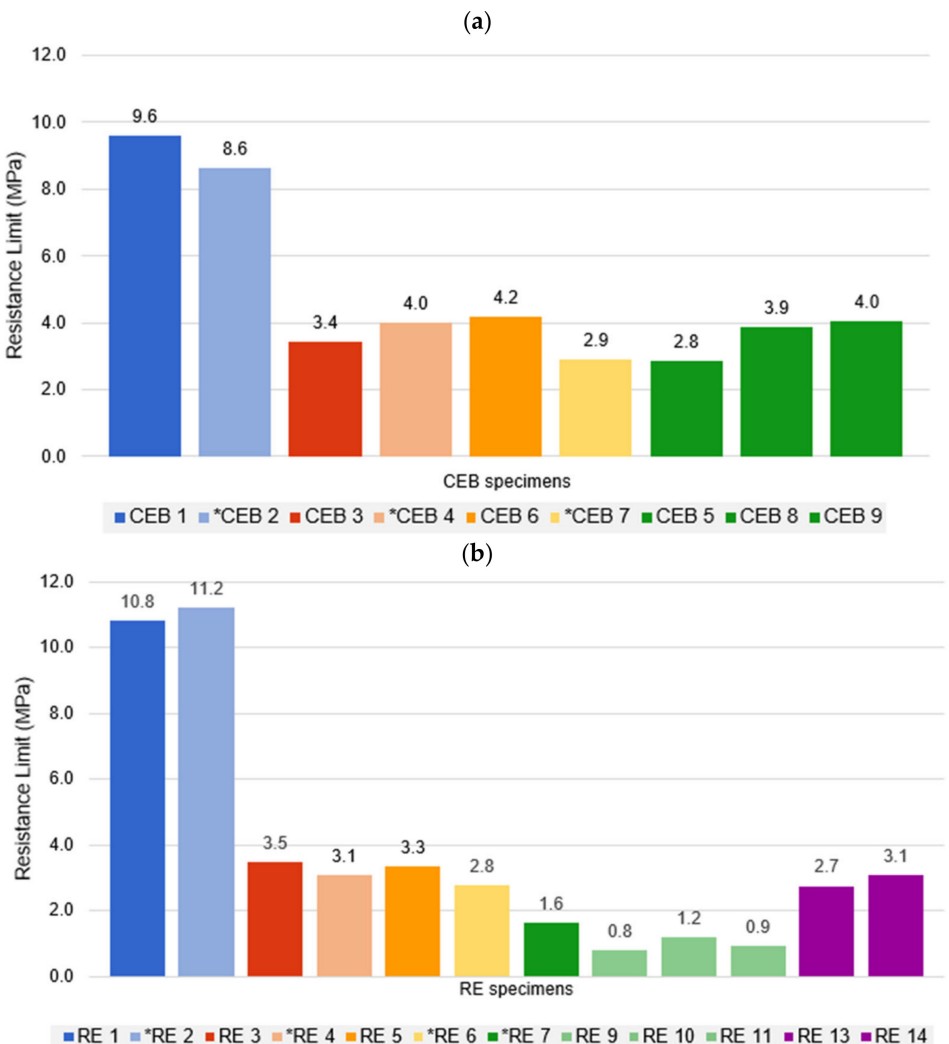

**Figure 13.** Compressive strength: (**a**) CEB specimens; (**b**) RE specimens (specimens marked with an asterisk were not tested for erosion).

According to the Spanish standard for compressed earth blocks [46], blocks with a compressive strength lower than 1.3 MPa are deemed unsuitable for construction purposes. Similarly, the Australian standard [47] requires a minimum value of 2 MPa for construction on rammed earth and CEBs. The tests conducted on the specimens produced in this study demonstrated good overall values of compressive strength, with stabilized specimens exhibiting results higher than 2.7 MPa. The unstabilised rammed-earth specimens that underwent erosion testing and were protected by lime mortar obtained compressive strength values ranging from 0.8 MPa to 1.2 MPa. The unstabilised specimen RE 7, which was without any protective layer and was not subjected to erosion testing, showed a compressive strength value of 1.6 MPa. A study conducted by Misseri et al. tested rammed-earth specimens as bi-modulus materials, consisting only of soil and water, which were subjected to uniaxial compression tests after curing, resulting in an average maximum stress of 2.567 MPa [48].

*5.4. Relation between Ultrasonic Analysis Results and Compressive Strength*

Through the ultrasonic test, it is possible to obtain information about the homogeneity of a material, namely, the presence of voids or the existence of cracks, in which materials that are less compacted have a lower wave propagation velocity than more compacted materials. It is also possible to relate, in a certain way, propagation velocity with the compressive strength of the material, according to Abreu [49].

In this work, a relationship was established between the variation in the ultrasonic test results before and after the wetting and drying cycles and the compressive strength assessed after the cycles. Contact with water causes the cohesion of the materials to improve, causing an increase in the wave velocity. This occurred with all specimens tested. However, a high increase in this velocity may indicate a loss of mass and a consequent loss of mechanical capacity. Tables 5 and 6 illustrate the relationship between the results of the ultrasonic velocity variation and their compressive strength for the CEB and RE specimens.

**Table 5.** Relationship between ultrasonic velocity variation and compressive strength for CEBs.

| Relationship between CEBs Results | | |
| --- | --- | --- |
| Specimens | Results | |
| Typology | Average transmission speed variation (%) | Compressive strength (MPa) |
| CEB 1 | 2.74 | 9.6 |
| CEB 3 | 15.37 | 3.4 |
| CEB 5 | 20.76 | 2.8 |
| CEB 6 | 8.09 | 4.2 |
| CEB 8 | 14.49 | 3.9 |
| CEB 9 | 12.69 | 4.0 |

**Table 6.** Relationship between ultrasonic and compression results of RE specimens.

| Relationship between the Results of RE Specimens | | |
| --- | --- | --- |
| Specimens | Results | |
| Typology | Average transmission speed variation (%) | Compressive strength (MPa) |
| RE 1 | 2.66 | 10.8 |
| RE 3 | 16.19 | 3.5 |
| RE 5 | 7.80 | 3.3 |
| RE 13 | 3.97 | 2.7 |
| RE 14 | 4.41 | 3.1 |

In Table 5 above, it is possible to observe the relationship between the increase in the wave transmission speed with the result of the mechanical compressive strength of the CEBs. Regarding the CEB with cement addition, which had a transmission speed variation of 2.74%, transmission speeds higher than 2000 m/s, and compressive strength index of 9.6 Mpa, better results were expected in relation to the samples with a lime addition. However, it is precisely in a comparison between the lime samples that it can be observed that the worst resistance index among the tested specimens was obtained by the CEB 5 specimen, which had the addition of quicklime and traditional varnish and obtained a greater wave transmission variation (20.76%) between the moment before the durability test and after the test. This specimen obtained a compressive strength value of 2.8 Mpa. It was observed in the other specimens with the addition of lime that the smaller the transmission variation measured before and after the durability test, the higher the index of compressive strength. In fact, better results were expected compared to CEB 5, requiring new evaluations of the traditional varnish by repeating these tests to make sure of their properties.

Similar to the CEBs, the ER specimen with the addition of cement performed better in relation to the other specimens with lime additions, in which the RE 1 specimen obtained a transmission speed variation of 2.66% before and after the durability test and a compressive strength index of 10.8 MPa. Among the ER samples with lime additions, this correlation

between the results of the two analyses was different. The RE 3 test piece, with the addition of hydrated lime, obtained a speed transmission variation of 16.19% and a compression strength index of 3.5 MPa, the best resistance index among the specimens with lime additions. However, specimen RE 13, with the addition of quicklime and a soaping finish, presented the lowest variation in transmission speed among the specimens with lime additions, demonstrating an increase of 3.97% of the velocity, though it was the specimen that also obtained the lowest index of compressive strength at 2.7 MPa. Therefore, to obtain more accurate results in this comparison, it is recommended that the tests be repeated with a larger number of samples of the same type. It is also recommended to increase the number of cycles and the drying time of the durability test.

## 6. Conclusions

Considering the level of education of the participants in the survey and the feasibility of access to information, the survey results revealed that in practice, major awareness and an actual change of paradigm regarding sustainability in construction have not yet occurred. The results also point out a predominance of illiteracy regarding traditional and sustainable construction materials, such as earthen construction, as well as their benefits. In general, designers considered the issue of sustainability to be relevant but did not give it due importance, and few put it into practice. Respondents also considered that their clients were particularly interested in the economic aspect, so they do not usually propose the use of more environmentally friendly construction materials and techniques. Therefore, there is a need for change in the understanding and acceptance of building techniques with natural materials since even some participants of the survey expressed their opinions that there is still a lack of collective awareness, engagement of professionals in the construction area, and stricter measures, such as legislation, that require the application of sustainable and ecological solutions.

Regarding the laboratory study, the rain simulation erosion test showed little loss of mass in cases where the soil was stabilized or where protective measures were used, with excellent results in most of the specimens. One of them showed zero variation, which proves that earthen construction can be durable, either by using a small percentage of stabilising material which, due to the wetting and drying cycles, promotes the reorganisation of particles and the formation of new structures, making the material more stable and resistant [42]; or by using a covering plaster layer. Apart from the samples that lacked additives but had mortar plaster, all other samples showed results that lie within the levels expected by international standards and are also compatible with the results of previous studies by other authors.

This study shows that it is simple to achieve soil improvement using low-cost materials, which makes it a safe and easy material to use in general construction, and it is also adaptable to a variety of climates.

**Author Contributions:** Conceptualization, J.F.N. and R.E.; methodology, J.F.N. and R.E.; software, J.F.N.; validation, J.F.N., R.E. and D.V.O.; formal analysis, R.E.; investigation, J.F.N., R.E. and D.V.O.; resources, R.E.; data curation, J.F.N.; writing—original draft preparation, J.F.N.; writing—review and editing, J.F.N., R.E. and D.V.O.; visualization, J.F.N., R.E. and D.V.O.; supervision, R.E. and D.V.O. All authors have read and agreed to the published version of the manuscript.

**Funding:** This research received no external funding.

**Data Availability Statement:** Not applicable.

**Conflicts of Interest:** The authors declare no conflict of interest.

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
