# Peer review of "Earthen Construction: Acceptance among Professionals and Experimental Durability Performance"

_constrmater, doi:10.3390/constrmater3020010_

Round 1

Reviewer 1 Report

The study employs combined qualitative and quantitative research methods to investigate the limitations and possibilities that stabilised rammed earth and compressed earth blocks might offer. The research is hence twofold and reports a direct survey on a mixed group of professional and non-professional subjects, and a series of mechanical and durability tests to show the effectiveness of the proposed admixtures.

The idea of combining outcomes from statistical/sociological analyses with evidence from laboratory tests to raise awareness and shed light on the possibilities that raw earth materials and technologies possibly offer is appealing.

It seems that the two aspects of the research can be bonded more efficaciously in the presentation section. The first two sections are both named Introduction, it might be useful to compact them into one.

To further support the structural and technological suitability of earthen constructions, and improve the reference list, a series of publications shall be inserted:

Fabbri, A., Morel, J-C. and Gallipoli, D. (2018) ‘Assessing the performance of earth building materials: a review of recent developments’, RILEM Technical Letters, December, Vol. 3,pp.46–58

Misseri, G., & Rovero, L. (2022). Rammed earth as bi-modulus material: Experimental and analytical investigations through Euler-Bernoulli and Timoshenko beam models. International Journal of Masonry Research and Innovation, 7(5), 482-503.

It would be interesting to know which questions were profiled (may be collected in an annexe or supplemental material), and in general to understand the number and the type (open/close, y/n, multiple choice, sections etc) to ease repeatability and scalability of the research. It would be interesting to insert some quantitative appraisal of the results of the survey using percentages and ratios.

Further minor changes:

Lines 153-157 - Please, provide descriptions before using acronyms (EPS, ETICS)

Line 159 – maybe the word “which” should be substituted with “who”.

Table 1, please include the number of specimens tested for each sample

Line 377-379, please fix the sentence. is it CEB5? or mass decrease?

Line 449 – fix typo red-bed.

Line 455 – Please refer first to the Figure number before commenting on it

Figure 9 – compressive strength values shall be accompanied by Coefficient of Variation values if the number of tested specimens is greater or equal to three.

Line 530 – Please replace “test pieces” with “specimens”

Reviewer 2 Report

The presented study, although interesting, requires the addressing of several issues. In addition to the following remarks, a file is attached with further comments.

1) The manuscript needs editing: language correction, missing commas, terminology etc. Some remarks are highlighted in the attached file but further effort is required.

2) There are two introductions, which should be merged into one. In the second introduction, the analysis of ancient materials is mentioned but not presented.

3) Section 3 could be smaller.

4) The conducted survey and the obtained results of section 4 are very interesting. The reviewer, however, is not sure they should be in the same paper with the laboratory research on rammed earth and CEB. If the authors decide to include the survey, it should be better presented. The results are presented in the text and it is difficult for the reader to follow. Tables and figures are essential to understand the survey and its results. Only one is presented in low resolution (Figure 1).

5) Section 5 could be a paper on its own. In this section there is not a materials and methods section as required by the journal. The materials and methods are presented in subsections 5.1 and 5.2, the latter of which is adequate in general, but important data are missing from 5.1. In lines 209-210 it is mentioned that geotechnical characterization was carried out on the soil used for the production of RE and CEB, yet this characterization is not presented. The properties of the soil used before the correction are of profound importance to understand the research and should be presented in detail. Furthermore, the other materials added are not discussed. The addition of Kaolin is mentioned, for example, but what kind of kaolin? With what properties Commercial, fine, coarse? What cement and what lime was added? Therefore, this section needs improvement.

Taking into consideration all the above, the reviewer proposes the major revision of the paper.

Round 2

Reviewer 2 Report

The paper is significantly improved. Especially section 4. Most of the concerns raised by the reviewer have been addressed. Those that weren’t are of minor importance except probably two: 

In line 281 the use of a mortar is mentioned with no further information. Is it an earth mortar, a cement mortar or what? 

The paragraph added in lines 491-506 is very useful and indeed the compressive strength testing of earth blocks is a matter of dispute. In this respect you could find useful and include in your paper DIN 18945 that distinguishes compressive strength testing of blocks according to their height; Those that their height is greater than 71 mm as yours, is suggested to be tested as single blocks. But this discussion about the testing procedure, and if you tested two blocks joined by mortar or single blocks, shouldn’t be at the results section but at section 5.2.3 where you describe the methods used. This is of great importance because a researcher that reads your paper would know if his results are directly comparable with yours or not. It also should be clearer in section 5.2.3 what you finally tested, single blocks or two blocks joined with mortar. 

Few textual oversights can also be corrected. 

In line 172 the % is missing after 13.6. 

In figure 13 in the caption the word specimens is underlined with a red line

Author Response

Dear Reviewer,

We would like to extend our sincere gratitude for your time and effort in reviewing our paper. Your valuable feedback has greatly contributed to improving the quality of our work.

We hope that all the issues and concerns you raised have been adequately addressed and that the revised manuscript now meets the high standards expected of it. Please see the attachment.

Thank you once again for your assistance in this process.

Best regards.
